

# Protein expression profile changes of lung tissue in patients with pulmonary hypertension

Min Wu, Yijin Wu, Jinsong Huang, Yueheng Wu, Hongmei Wu, Benyuan Jiang and Jian Zhuang

Department of Cardiac Surgery, Guangdong Provincial People's Hospital, Guangzhou City, Guangdong Province, China

Corresponding authors
Min Wu, wumin0011@gdph.org.cn
Jian Zhuang,
zhuangjian5413@tom.com

## ABSTRACT

**Background**. Pulmonary hypertension occurs in approximately 1% of the global population, and the prognosis for such patients may be poor. However, the mechanisms underlying the development of this disease remain unclear. Thus, understanding the development of pulmonary hypertension and finding new therapeutic targets and approaches are important for improved clinical outcomes.

**Methods**. Lung tissue specimens were collected from six patients with atrial septal defect and pulmonary hypertension (all women, with a mean age of $46.5 \pm 4.7$ years, and their condition could not be corrected with an internal medical occlusion device) and from nine control patients with lung cancer who underwent lobectomy (six men and three women, with a mean age of $56.7 \pm 1.7$ years). Isobaric tags for relative and absolute quantitation and liquid chromatography tandem mass spectrometry analyses were used to detect protein expression levels.

**Results**. We found 74 significantly upregulated and 88 significantly downregulated differentially expressed proteins between control and pulmonary hypertensive lung tissue specimens. Gene ontology analyses identified the top 20 terms in all three categories, that is, biological process, cellular component, and molecular function. Kyoto Encyclopedia of Genes and Genomes and protein–protein interaction analyses determined the top 10 signaling pathways and found that the six hub proteins associated with the differentially expressed upregulated proteins (PRKAA1, DHPR, ACTB, desmin, ACTG1, and ITGA1) were all involved in hypertrophic cardiomyopathy, arrhythmogenic right ventricular cardiomyopathy, and dilated cardiomyopathy.

**Conclusion**. Our results identified protein expression profile changes in lung tissue derived from patients with pulmonary hypertension, providing potential new biomarkers for clinical diagnosis and prognosis for patients with pulmonary hypertension and offering candidate protein targets for future therapeutic drug development.

## INTRODUCTION

Pulmonary hypertension is a chronic, persistent complex disease involving many lesions (*Kim & George, 2019*). The incidence of pulmonary hypertension is approximately 1% of the global population (*Hoeper et al., 2016*). Epidemiological surveys in Europe and

the United States show that idiopathic pulmonary hypertension is most common in women and that familial pulmonary hypertension is associated with genes (*Franco, Ryan & McLaughlin, 2019*; *McGoon et al., 2013*). Pulmonary hypertension is a chromosomal dominant genetic disease mainly attributable to bone morphogenetic protein receptor-2 mutations (*Harper et al., 2019*). Other types of pulmonary hypertension are often associated with a variety of diseases, including congenital heart disease, chronic obstructive pulmonary disease, connective tissue disease, drug and toxin effects, HIV/AIDS, hemoglobinopathy, and coagulation disorders (*Basyal, Jarrett & Barnett, 2019*; *Kim & George, 2019*; *Mitra et al., 2018*; *Zhang et al., 2019*). Pulmonary hypertension can also be related to certain atmospheric conditions, such as hypoxia observed at high altitude. Without effective treatment, the prognosis for patients with pulmonary hypertensive is generally poor. The annual mortality rate of such patients is approximately 15% (*Thenappan et al., 2007*). Poor cardiopulmonary function, low mobility, increased right atrial pressure, progressive right ventricular failure, low cardiac output, increased brain natriuretic peptide, and progression of connective tissue disease are all predictors of poor prognosis (*Fukuda et al., 2019*).

Recent studies have revealed several new candidate targets and approaches for development of future pulmonary hypertension treatment. *Zhang et al. (2019)* have suggested that autophagy of pulmonary artery endothelial and smooth muscle cells may induce dysfunction of the pulmonary arteries and eventually lead to pulmonary hypertension (*Zhang et al., 2019*). *Lambert et al. (2018)* have reported that ion channel activity changes in pulmonary artery endothelial and smooth muscle cells may pathophysiologically contribute to the development of pulmonary hypertension (*Lambert et al., 2018*). Additionally, immune system dysfunction and inflammation may be linked with pulmonary arterial hypertension pathogenesis (*Goldenberg & Steinberg, 2019*). Therefore, it is important to find new and effective targets to treat pulmonary hypertension.

Isobaric tags for relative and absolute quantitation (iTRAQ) in combination with liquid chromatography tandem mass spectrometry (LC–MS/MS) are powerful tools for identifying protein expression levels of the various types of proteins that are in a specimen. In the present study, we obtained pulmonary tissue specimens from patients undergoing surgical pulmonary procedures and used the iTRAQ and LC–MS/MS methods to identify proteins that were differentially expressed in the pulmonary tissue of patients with pulmonary hypertension versus those from patients without this disease. Using bioinformatics analysis, we assessed those differentially expressed proteins (DEPs) to determine potential key proteins and their related signaling pathways that may be associated with the development of pulmonary hypertension.

## MATERIALS & METHODS

### Clinical specimens and patient enrollment

Our research plan and an informed patient consent form for the present study were submitted to the Medical Research Ethics Committee of Guangdong Provincial People's Hospital. The Medical Ethics Committee reviewed and approved our submitted proposal,

ethical approval number Guangdong Medical Ethics 2016220H (R1)/GDREC2016220H (R1). All patients signed the written informed consent form prior to being included in the study.

Patients were enrolled in strict accordance with the following inclusion and exclusion criteria, as specified in the experimental protocol. For the experimental group, the included patients had been diagnosed as having pulmonary hypertension with an atrial septal defect that could not be corrected by internal closure. In the resting state, with the patient lying so that the left atrium was in a horizontal position and the measurement determined at a distance midway between the middle of the sternum and the bedside, the average pressure of the pulmonary artery measured by a catheter located in the right ventricle of the heart was $\geq 25$ mmHg. For the control group, the included patients did not have pulmonary hypertension and had received a diagnosis of central lung cancer requiring lobectomy, but the cancer tissue had not invaded the peripheral lung tissue used in this study.

## iTRAQ labeling and fractionation by cation exchange chromatography

The 8-plex iTRAQ labeling method was performed as previously described (*Ding et al., 2019*; *Liu et al., 2019*). The reagents were used to label protein peptides from each group multiplex kit (ABI, Foster City, CA, USA) (isobaric tags 113, 114, and 116 for the control group, and isobaric tags 118, 119, and 121 for the pulmonary hypertension group). For each specimen, protein (200 μg) was precipitated with acetone at $-20\,°C$ overnight. After centrifugation for 10 min, the protein pellet was dissolved in 60 μL of iTRAQ dissolution buffer (Applied Biosystems). The iTRAQ labeling reagents were added to the corresponding peptide specimen to react at room temperature for 1 h. After the reaction for the labeling was stopped and an extraction efficiency test was performed, the specimen was ZipTip desalted and subjected to a matrix-assisted laser desorption ionization procedure. Each group was pooled and then vacuum-dried. Each pooled specimen containing mixed peptides was lyophilized and dissolved in solution A (2% acetonitrile and 20 mM ammonium formate, pH 10). This solution was loaded on a reverse-phase column (Luna C18, $4.6 \times 150$ mm; Phenomenex; Torrance, CA, USA) and eluted using a step linear elution program at a flow rate of 0.8 mL/min: 0%–10% buffer B (500 mM KCl, 10 mM $KH_2PO_4$ in 25% acetonitrile, pH 2.7) for 10 min, 10%–20% buffer B for 25 min, 20%–45% buffer B for 5 min, and 50%–100% buffer B for 5 min. The eluent fractions were collected and centrifuged for 5–45 min. The fractions were finally combined into six pools and desalted using C18 Cartridges (Empore standard density Solid Phase Extraction C18 Cartridges, bed I.D. seven mm, three mL volume; Sigma; St. Louis, MO, USA).

## LC–MS/MS analysis

We performed MS using a TripleTOF 5600 system (AB SCIEX) combined with a nanoliter spray III ion source (AB SCIEX, USA). The spray voltage was set at 2.5 kV; the air curtain pressure, at 30 psi; the atomization pressure at 5 psi; and the heater temperature, at 150 °C. The scanning mode used was information-dependent acquisition. The scan time of the first time-of-flight (TOF)–MS single image was 250 ms. Each information-dependent
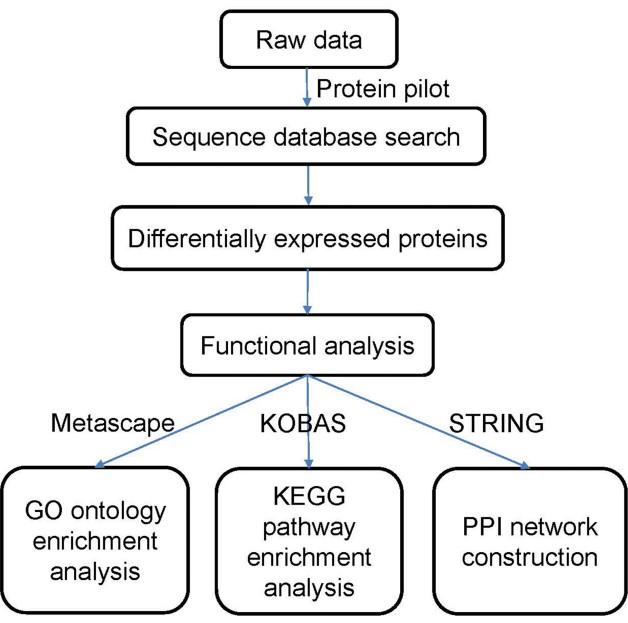

**Figure 1  Flowchart of the data analysis procedure.**

acquisition cycle collected 35 secondary TOF–MS images. Each cycle time was fixed at 2.5 s. The dynamic exclusion was set to 18 s, which was approximately equal to half the width of the chromatogram.

## Sequence database search and data analysis

The data were processed according to the flowchart shown in Fig. 1. After the data were collected, we used Protein Pilot Software v. 5.0 (AB SCIEX, USA) for the analysis. Human proteome databases containing reviewed Uniprot sequences were used to perform peptide identification. Decoys for the database search were generated with the revert function. We chose the following options to identify the proteins: cysteine alkylation = iodoacetamide; enzyme = trypsin; search effort = thorough. The proteins that showed a relative upregulated or downregulated fold change of at least 1.2 (in both replicates) and that were significantly different between the groups at $P <0.05$ were determined to be differentially expressed between the groups.

## Gene ontology (GO) and Kyoto Encyclopedia of Genes and Genomes (KEGG) pathway enrichment analyses

We performed GO analyses using Metascape, a web-based resource for gene and protein annotation, visualization, and integration discovery (http://metascape.org) (*Fang et al., 2019*; *Soonthornvacharin et al., 2017*). KEGG pathway analyses of the DEPs were performed using the KOBAS online analysis database (http://kobas.cbi.pku.edu.cn/) (*Kanehisa & Goto, 2000*). A two-sided $P <0.05$ was considered statistically significant.

**Table 1  Results of protein identification analyses.** Using an Unused ProtScore cutoff >1.3 and peptides ≥1, we obtained 2,953 proteins.

| Unused ProtScore (Conf) cutoff | Proteins detected, No. | Proteins before grouping, No. | Distinct peptides, No. | Spectra identified, No. | % of Total spectra |
|---|---|---|---|---|---|
| >2.0 (99) | 2,502 | 8,613 | 35,550 | 119,806 | 46.2 |
| >1.3 (95) | 3,118 | 12,147 | 36,406 | 121,271 | 46.8 |
| >0.47 (66) | 3,251 | 12,682 | 36,614 | 121,540 | 46.9 |
| Cutoff >0.05 (10%) applied | 3,497 | 14,210 | 37,010 | 122,026 | 47.1 |

## Protein–protein interaction (PPI) network construction and module analyses

PPI analysis was used to assess the functions associated with the DEPs and to determine the general organizational principles of the functional cellular networks. The functional relationships between proteins were identified using the Search Tool for the Retrieval of Interacting Genes (STRING; http://string.embl.de/) (*Fang et al., 2019*; *Von Mering et al., 2003*). The PPI networks associated with the respective DEPs were constructed to predict the interaction of those proteins.

# RESULTS

## Specimen collection and patient characteristics

From April to September 2016, we collected lung tissue specimens from six patients with atrial septal defect and pulmonary hypertension. The patients were all women, with a mean (SD) age of $46.5 \pm 4.7$ years, and their condition could not be corrected with an internal medical occlusion device. For the control group, nine patients with lung cancer who underwent lobectomy were enrolled: six men and three women, with a mean (SD) age of $56.7 \pm 1.7$ years. Among the patients with pulmonary hypertension, the mean pulmonary systolic pressure was $66.7 \pm 5.5$ mmHg; mean pulmonary artery diastolic pressure, $23.2 \pm 1.6$ mmHg; mean pulmonary artery pressure, $39.7 \pm 3.4$ mmHg; mean pulmonary artery resistance, $3.27 \pm 0.28$ Wood units; pulmonary oxygen saturation, $87.3\% \pm 0.84\%$; and mean patient body surface, $1.52 \pm 0.06$ m$^2$.

## Identification of DEPs in pulmonary tissue

After performing the quantitative analyses with Protein Pilot Software, we identified 2,953 proteins using a detection protein threshold (Unused ProtScore [Conf]) cutoff of >1.3; 95% confidential interval) (Table 1), and we obtained the protein expression fold changes between the control specimens and the pulmonary hypertensive lung tissue specimens. The data indicated marked protein expression profile changes between the control lung tissue and the pulmonary hypertensive lung tissue. Using our selection criteria, we identified 74 significantly upregulated DEPs, including integrin subunit α1 (ITGA1) and the voltage-gated calcium ($Ca^{2+}$) channel α2/ δ1 subunit, and 88 significantly downregulated DEPs, including Rho-GDP dissociation inhibitor 2 (Table 2, Fig. 2).

**Table 2  Fold changes of key proteins in pulmonary hypertensive lung tissue.**

| Protein ID (UniProt) | Gene symbol | Fold change |
|---|---|---|
| P54289 | CACNA2D1 | 1.246 |
| P17661 | DES | 1.257 |
| Q13131 | PRKAA1 | 1.794 |
| P60709 | ACTB | 1.215 |
| P56199 | ITGA1 | 1.209 |
| P63261 | ACTG1 | 1.215 |
| P52566 | ARHGDIB | 0.743 |

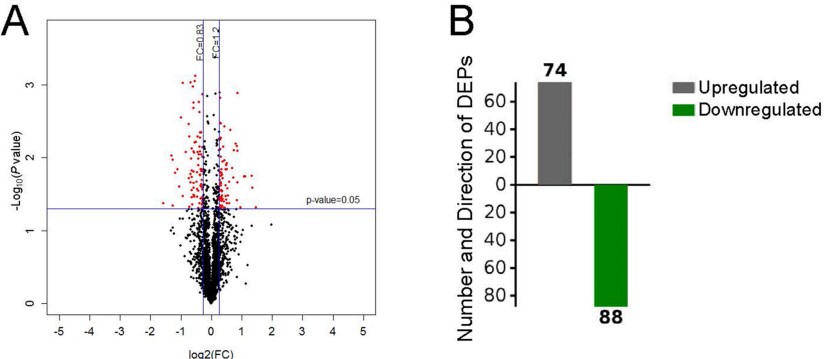

**Figure 1  Differential protein expression.** (A) Volcano plot with red dots on the right-hand side indicating upregulation, red points on the left-hand side indicating downregulation, and black dots indicating no significant change in protein expression levels based on the criteria of an absolute log2 fold change (FC) >1.2 and $P < 0.05$. (B) Bar graph indicating 74 upregulated and 88 downregulated proteins. DEP indicates differentially expressed protein.

## GO functional enrichment analysis

GO analysis is used to hierarchically classify genes or gene products into categories organized in an ontology. The analysis is based on three categories: (1) *molecular function*, to describe the molecular activity of a gene; (2) *biological process*, to characterize the larger cellular or physiological role of the gene; and (3) *cellular component*, to indicate the location of the gene or gene product in the cell. We used Metascape to enrich all identified proteins. Among the top 20 terms in the biological process category, the highest percentages of the proteins were associated with the term *single-organism process* (Figs. 3 and 4) ($n = 119$ proteins; with the top 3 upregulated proteins in this process being LDLRAP1, FTL, and CHUK). The maximum levels of the expressed proteins this category were detected for the term *protein complex subunit organization* ($n = 27$ proteins; with the top 3 upregulated proteins being SLC6A4, FRYL, and HIST2H3A). Among the top 20 terms in the GO category cellular component, the highest percentages of proteins were associated with the term *cytoplasm* ($n = 104$ proteins, with the top 3 upregulated proteins for this term being LDLRAP1, FTL and CHUK. The maximum levels of the expressed proteins in this category were in the term *major histocompatibility complex* (MHC), which is a group of

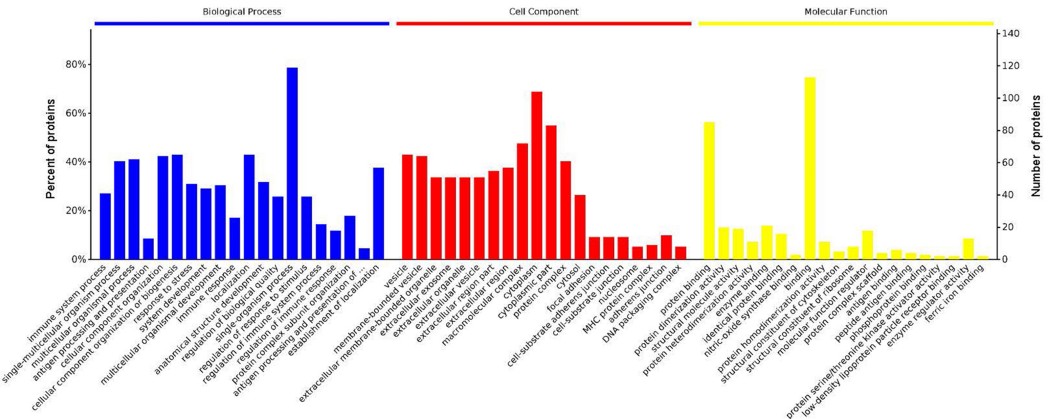

**Figure 3** **Enrichment of the gene ontology functional terms in the top 20 entries of the three categories of biological process, cellular component, and molecular function for highly expressed proteins.** The entries in each category are sorted from left to right according to their −log P value. The more significant, and the number of proteins corresponding to each item, can be viewed from the left to right vertical axes. The number of proteins is a percentage of the total number of submitted proteins. For the category biological process, the full name of "antigen processing and presentation of . . ." is antigen processing and presentation of peptide or polysaccharide antigen via MHC class II.

proteins located on the cell surface ($n = 9$ proteins, with the top 2 upregulated proteins being HLA-A and HLA-DPA1). In the top 20 terms of the Go category molecular function, the highest percentages of proteins were associated with the term *binding* (113 proteins, with the top 3 upregulated proteins being LDLRAP1, FTL and CHUK), and the maximum levels of the expressed proteins in this category were for the term *ferric iron binding* (with the top 2 upregulated proteins being FTL and FTH1). The GO functional enrichment analysis assessing the distributions of the proteins with the greatest fold changes in the biological process, molecular function, and cellular component categories indicated that the proteins associated with the terms *organism process, protein binding*, and *extracellular vesicle*, respectively, were markedly changed.

## KEGG signaling pathway analysis and PPI network construction

To identify the functions associated with the DEPs, we used the KOBAS online analysis tool to determine the KEGG signaling pathways, and we used the STRING online analysis tool to construct the PPI networks. Through KEGG signaling pathway analysis, the gene content in the provided genome is compared with the information in the KEGG pathway database to clarify which pathways and associated functions link to the genes in the genome. The PPI network indicates physical contacts between two or more proteins resulting from biochemical events controlled by electrostatic forces, including hydrophobic effects. The top 10 KEGG signaling pathways were enriched, and the PPI networks were constructed (Figs. 5–9). We found that the top 10 signaling pathways included hypertrophic cardiomyopathy, systemic lupus erythematosus, arrhythmogenic right ventricular cardiomyopathy, dilated cardiomyopathy, pathogenic *Escherichia coli* infection, viral myocarditis, phagosome, alcoholism, cardiac muscle contraction, and

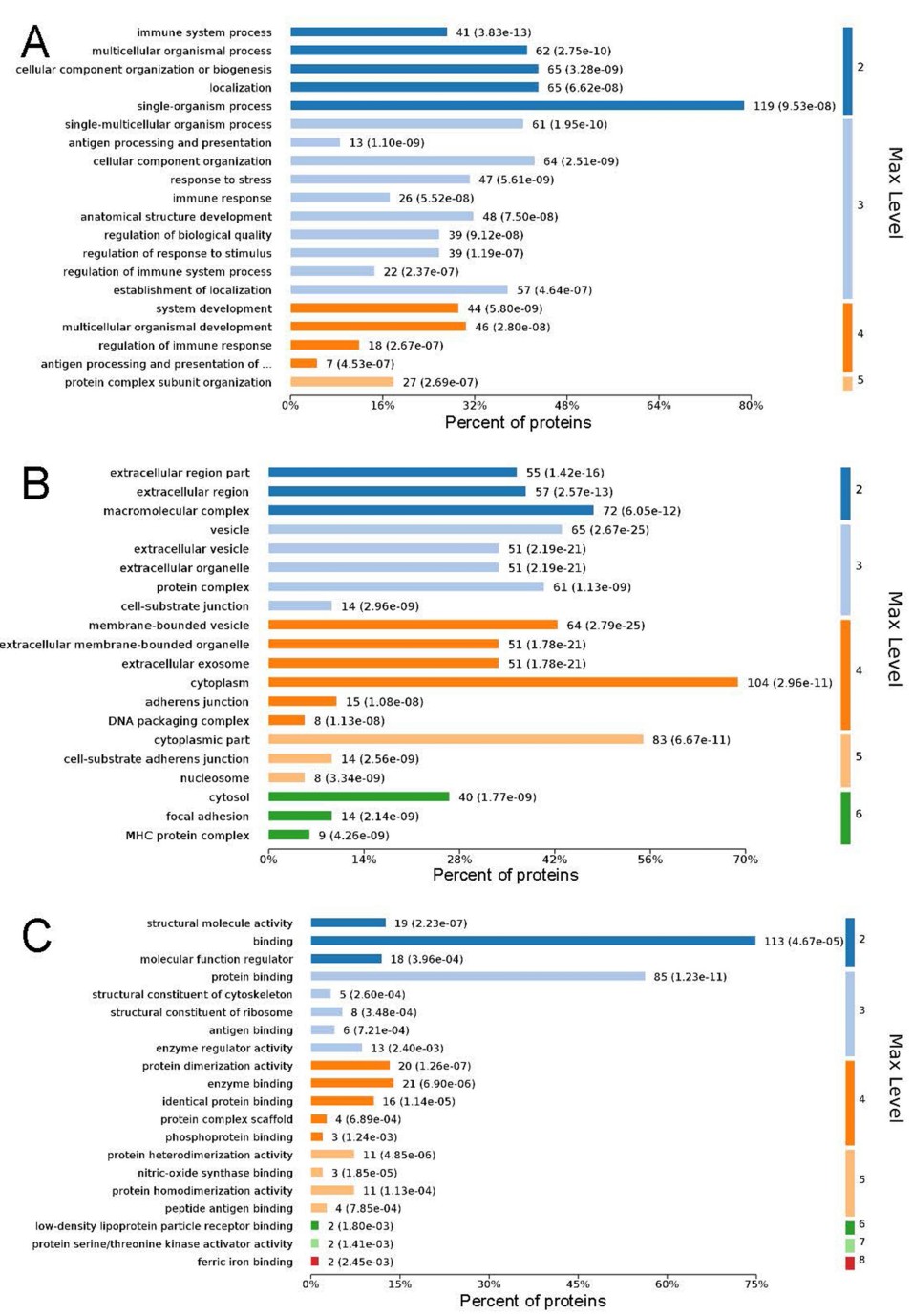

**Figure 4   Maximum level corresponding to the top 20 proteins in the gene ontology enrichment categories of biological process (A), cellular component (B), and molecular function (C), and the percentages of enriched proteins.** The ordinate is arranged from low to high according to the maximum level, and each level is arranged by *P* value. The abscissa is the percentages of the enriched proteins. The number at the end of each bar is the number of proteins enriched in the biological process, cellular component, or molecular function gene ontology category. In A, the full name of "antigen processing and presentation of …" is antigen processing and presentation of peptide or polysaccharide antigen via MHC class II.

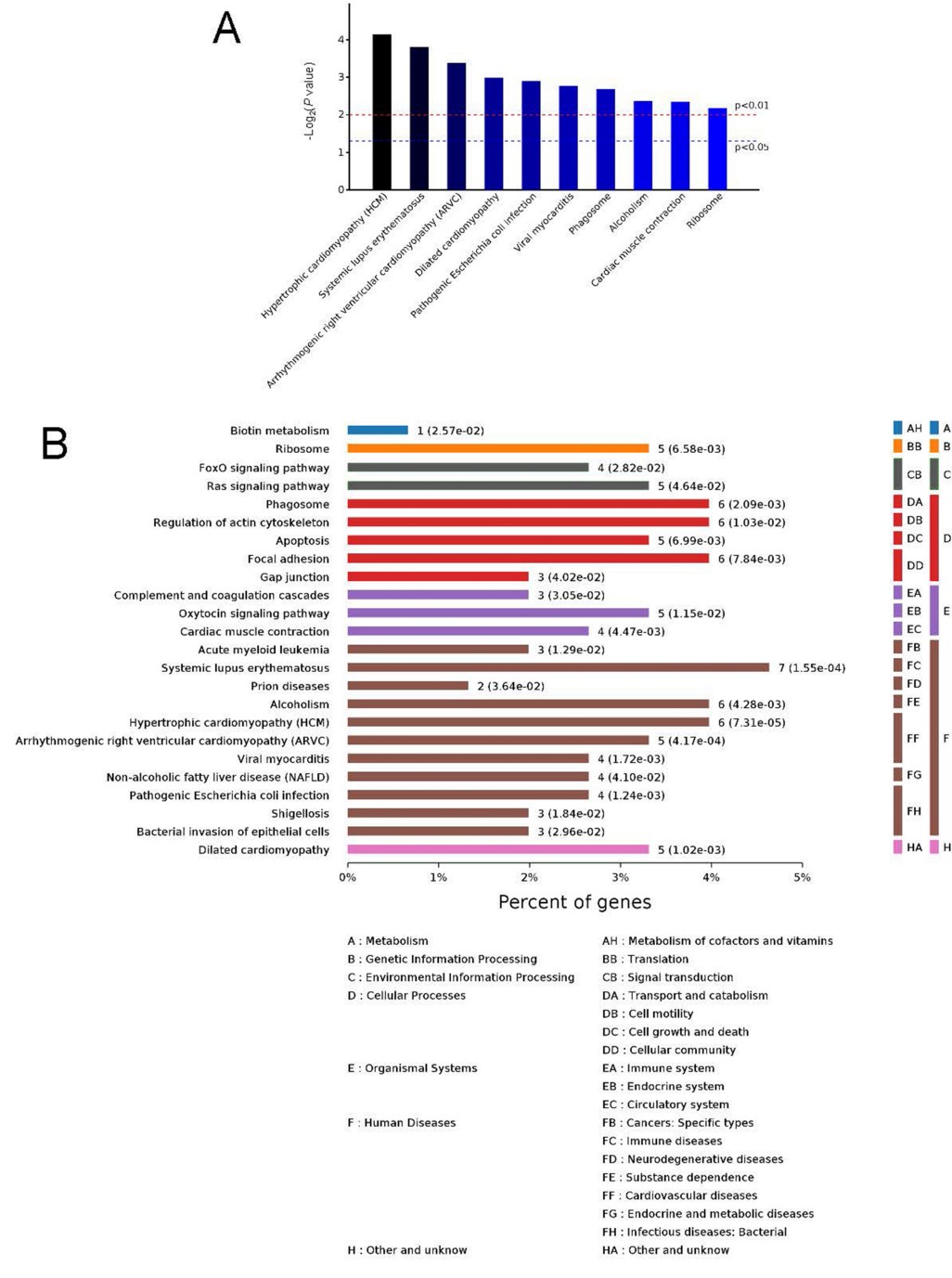

**Figure 5 Kyoto Encyclopedia of Genes and Genomes (KEGG) signaling pathways.** Top 10 KEGG signaling pathways (A) and KEGG classification histogram (B). In B, all enriched pathways with significant differences in *P* values are shown. The ordinate is the specific pathway classification and name. On the right, the number of associated genes, *P* values, and their corresponding KEGG classifications are shown. The abscissa gives the percentages of the genes.

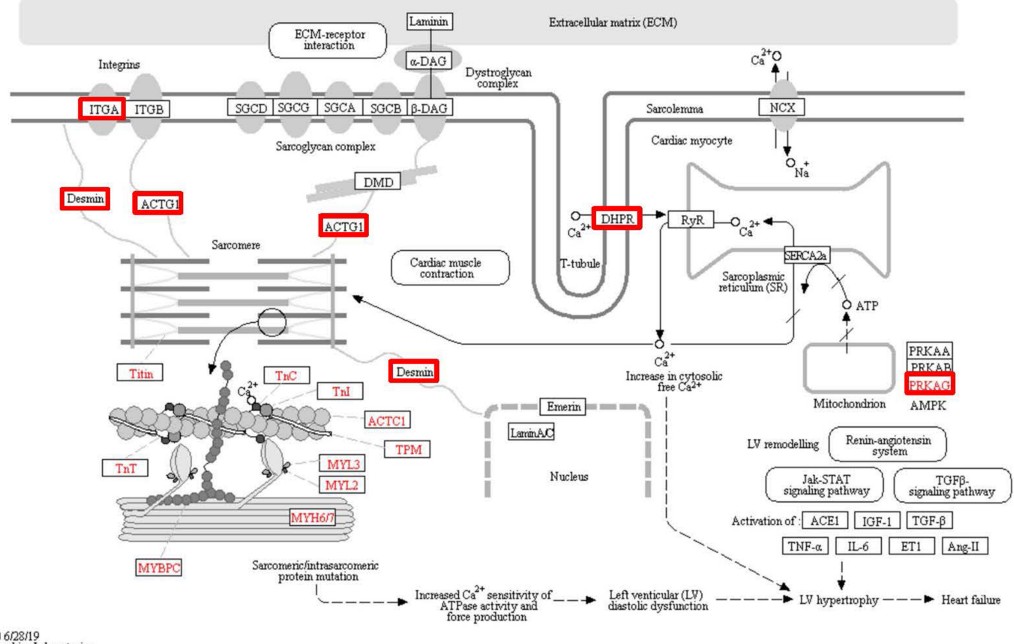

**Figure 6 Changes in the hypertrophic cardiomyopathy signaling pathway in lung tissue of patients with pulmonary hypertension.** Upregulated proteins are highlighted in red boxes.

ribosomes, which are predominantly associated with human diseases (Fig. 5). Some of the DEPs crossed several different signaling pathways. For example, the proteins highlighted in red boxes in Figs. 6, 7 and 8 including DHPR, ACTB, desmin, ACTG1, PRKAA1 and ITGA1, were all involved in hypertrophic cardiomyopathy, arrhythmogenic right ventricular cardiomyopathy, and dilated cardiomyopathy (Table 2). The results of the KEGG pathway enrichment and PPI network construction analyses indicated that many of the upregulated proteins (shown as brown dots in Fig. 9) were involved in hypertrophic cardiomyopathy, By contrast, many of the downregulated proteins (shown as green dots in Fig. 9) were related to ribosome function. Taken together, our results suggest that the heart is a major organ impaired by pulmonary hypertension.

## DISCUSSION

In the present study, we used clinical specimens obtained from patients undergoing surgical procedures, and we used the iTRAQ method combined with LC–MS/MS to identify the key proteins and signaling pathways associated with the development of pulmonary hypertension. Our primary findings were as follows: (1) We identified 2953 proteins, including 74 significantly upregulated DEPs and 88 significantly downregulated DEPs, between control and pulmonary hypertensive lung tissues; (2) GO analysis elucidated the top 20 terms associated with categories of biological process, cellular component, and molecular function; (3) KEGG and PPI analyses identified the top 10 signaling pathways and

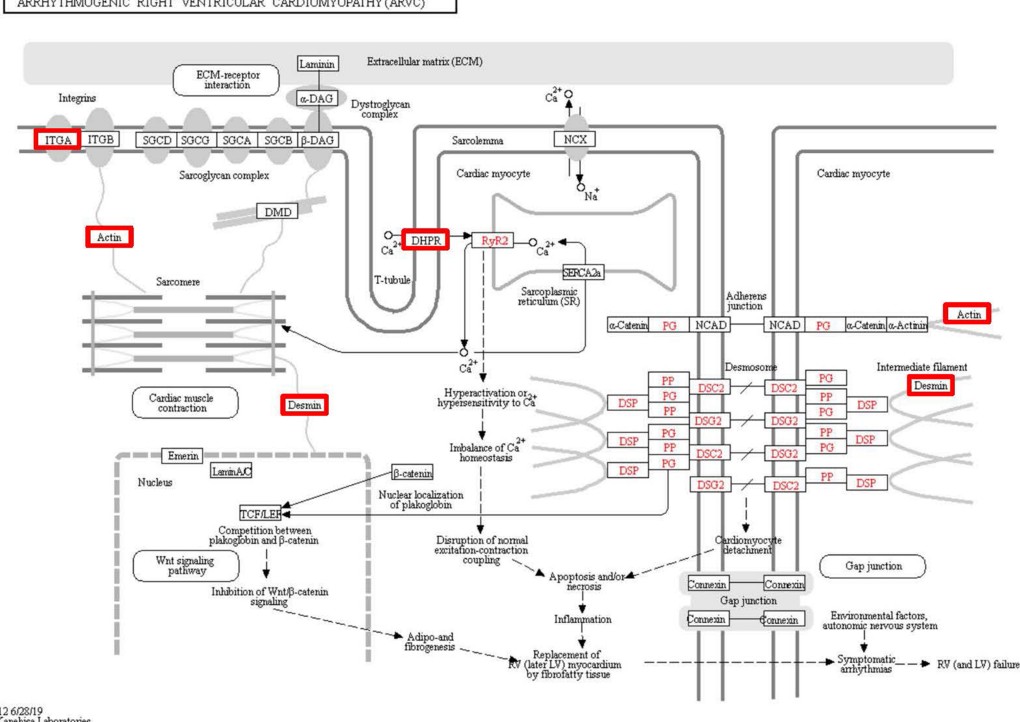

**Figure 7** **Changes in the arrhythmogenic right ventricular cardiomyopathy signaling pathway in lung tissue of patients with pulmonary hypertension.** Upregulated proteins are highlighted in red boxes.

six hub proteins, including PRKAA1, DHPR, ACTB, desmin, ACTG1 and ITGA1. Taken together, our study revealed protein expression profile changes in lung tissue of patients with pulmonary hypertension, providing a deeper understanding of the development of pulmonary hypertension and suggesting several potential targets for the development of new drugs in the treatment of pulmonary hypertension.

In the present study, we detected a high expression of *PRKAA1*, which encodes 5′-AMP-activated protein kinase catalytic subunit alpha-1 (AMPKα1). AMPKα1 is an enzyme involved in the pathological changes of smooth muscle cells in vessels of patients with pulmonary hypertension. AMPKα1 can maintain smooth muscle cells and their survival in an anoxic environment, and inhibition of AMPKα1 leads to hypoxia-induced activation of autophagy. In a hypoxic environment, AMPK is activated and phosphorylated in smooth muscle cells. Under hypoxic conditions, lactate dehydrogenase activity is increased after AMPK activity is inhibited by the AMPK antagonist compound C, leading to the induction of apoptosis in smooth muscle cells (*Ibe et al., 2013*). *Evans et al. (2015)* have suggested that the levels of AMPK and liver kinase B1 (LKB1) are upregulated when mitochondria sense a change in the oxygen supply. These upregulated levels cause a sustained blood vessel contraction through an LKB1–AMPK signaling pathway and result in continuous pulmonary hypertension. AMPK is activated by an increase in the ratio of ADP to ATP and rapidly amplifies the signaling pathway in synergy with LKB1, maintaining the intracellular

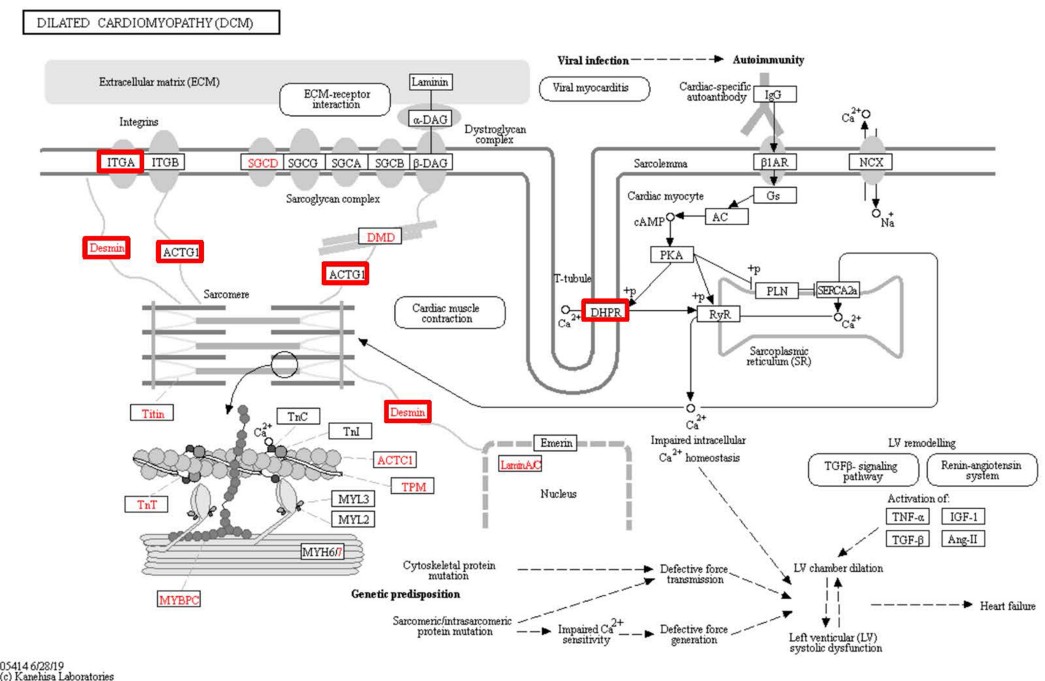

**Figure 8** **Changes in the viral myocarditis signaling pathway in lung tissue of patients with pulmonary hypertension.** Upregulated proteins are highlighted in red boxes.

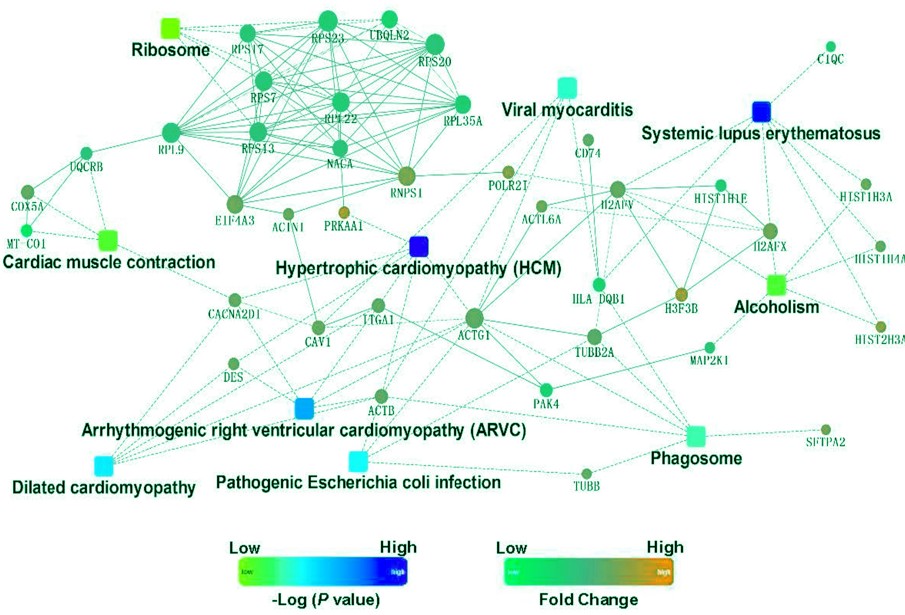

**Figure 9** **Protein–protein interaction networks in the Kyoto Encyclopedia of Genes and Genomes pathways that are ranked as the top 10 regulatory pathways.**

ATP concentration by catabolism and inhibition of unnecessary ATP depletion. Therefore, AMPK regulates the mitochondrial energy balance and maintains the energy supply of smooth muscle cells during hypoxia to preserve normal physiological functions (*Evans et al., 2015*). In the present study, the expression of AMPKα1 was significantly increased in the lung tissue of patients with pulmonary hypertension. This finding suggests that AMPK may be a potential target in the treatment of pulmonary hypertension.

We also found an upregulation of ITGA1, which is involved in the regulation of phosphorylation and the biological processes of essential metabolic processes. The integrins α1–5, α7–8, αv, β1, β3, and β4 have been detected on pulmonary vessels. Umesh et al. used a mouse model of chronic hypoxic pulmonary hypertension and performed immunohistochemical staining of pulmonary artery smooth cells to find that the level of ITGA1 expression on the cell surface was significantly increased, whereas that of integrin subunit α5 was significantly decreased (*Umesh et al., 2011*; *Umesh et al., 2006*). These expression level changes were mainly concentrated on the smooth muscle cells of peripheral pulmonary arterioles and were associated with $Ca^{2+}$ flow in smooth muscle cells (*Umesh et al., 2011*). Eventually, a $Ca^{2+}$ concentration change in vascular smooth muscle cells affects blood vessel contraction and blood pressure. In the present study, we also found an increase in ITGA1 in the lung tissue of patients with pulmonary hypertension. Thus, the increased level of this integrin may be a key reason for the development of pulmonary hypertension.

Smooth muscle cell contraction is regulated by intracellular $Ca^{2+}$ and Rho kinase signaling pathways (*Berridge, 2008*). $Ca^{2+}$ channels are involved in the pathological processes associated with pulmonary hypertension. Thus, $Ca^{2+}$ channel antagonists have been used in the early stages of clinical treatment of pulmonary hypertension, and some patients have achieved some therapeutic effects (*Kennedy, Michael & Summer, 1985*; *Packer, 1985*; *Packer, Medina & Yushak, 1984*; *Rubin, 1985*). In the present study, we found that the voltage-gated $Ca^{2+}$ channel subunit α2/δ1 encoded by *CACN2D1* significantly increased in the lung tissue of patients with pulmonary hypertension. Moreover, we also found that the *ARHGDIB*-encoded Rho-GDP dissociation inhibitor 2 protein is downregulated (Table 2). Rho-GDP dissociation inhibitor 2 can inhibit the dissociation of GDP from Rho protein, thereby regulating the rate of the GDP/GTP exchange reaction and inhibiting smooth muscle cell contraction (*Adra et al., 1993*; *Scherle, Behrens & Staudt, 1993*). Therefore, an increase in voltage-gated $Ca^{2+}$ channels and a decrease in Rho-GDP dissociation inhibitor 2 levels will eventually induce smooth muscle cell contraction, increase vascular resistance, and participate in the pathogenesis of pulmonary hypertension.

In the present study, we also found that three muscle contraction–related proteins, including β-actin encoded by *ACTB*, γ-actin encoded by *ACTG1*, and desmin encoded by *DES*, were significantly increased. Beta-actin is one of two nonmuscle cytoskeletal actins, and it is involved in cell motility, structure, and integrity (*Gunning et al., 2015*). Alpha-actins are a major constituent of the contractile apparatus (*Yonemura, 2017*). In adult striated muscle cells, γ-actin localizes to Z-disks and costamere structures and is involved in force transduction and transmission in muscle cells (*Nakata, Nishina & Yorifuji, 2001*; *Papponen et al., 2009*). Desmin is a type III intermediate filament in cardiac

muscle, skeletal muscle, and smooth muscle tissue. It integrates the sarcolemma, Z disk, and nuclear membrane in sarcomeres and regulates sarcomere architecture (*Brodehl, Gaertner-Rommel & Milting, 2018*; *Sequeira et al., 2014*). Therefore, these upregulated muscle contraction–related proteins may enhance vascular smooth muscle cell contraction to increase vascular resistance and may thus be involved in the pathogenesis of pulmonary hypertension.

## CONCLUSIONS

We used the iTRAQ method to identify 2953 proteins, including 74 significantly upregulated DEPs and 88 significantly downregulated DEPs between control and pulmonary hypertensive lung tissues. Through bioinformatics analyses, we found six increased key proteins, including PRKAA1, DHPR, ACTB, desmin, ACTG1, and ITGA1. The findings of the present study provide potential new biomarkers for clinical diagnosis and prognosis for pulmonary hypertension and candidate protein targets for drug development.

### Funding
This work was supported by grants from the Guangdong Project of Science and Technology (2017A070701013, 2017B090904034, 2017B030314109). The funders had no role in study design, data collection and analysis, decision to publish, or preparation of the manuscript.

### Grant Disclosures
The following grant information was disclosed by the authors:
Guangdong Project of Science and Technology: 2017A070701013, 2017B090904034, 2017B030314109.

### Competing Interests
The authors declare there are no competing interests.

### Author Contributions
- Min Wu conceived and designed the experiments, performed the experiments, analyzed the data, prepared figures and/or tables, authored or reviewed drafts of the paper, and approved the final draft.
- Yijin Wu and Jinsong Huang performed the experiments, analyzed the data, authored or reviewed drafts of the paper, and approved the final draft.
- Yueheng Wu analyzed the data, prepared figures and/or tables, authored or reviewed drafts of the paper, and approved the final draft.
- Hongmei Wu analyzed the data, prepared figures and/or tables, and approved the final draft.
- Benyuan Jiang conceived and designed the experiments, prepared figures and/or tables, and approved the final draft.
- Jian Zhuang conceived and designed the experiments, prepared figures and/or tables, authored or reviewed drafts of the paper, and approved the final draft.
## Human Ethics

The following information was supplied relating to ethical approvals (i.e., approving body and any reference numbers):

The Medical Research Ethics Committee of Guangdong Provincial People's Hospital (Guangdong Academy of Medical Sciences) approved this research (016220H (R1)/GDREC2016220H (R1)).

## Data Availability

The raw measurements are available in the Supplementary Files.

## Supplemental Information

Supplemental information for this article can be found online at http://dx.doi.org/10.7717/peerj.8153#supplemental-information.

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
