# Peer review of "Protein expression profile changes of lung tissue in patients with pulmonary hypertension"

_PeerJ, doi:10.7717/peerj.8153_

## Round 0.1 · original submission · Major Revisions

All critiques of all reviewers have to be carefully addressed and the manuscript should be revised accordingly.

Reviewer 1 ·

Basic reporting

The authors claim they found 74 significantly upregulated and 88 significantly downregulated differentially expressed proteins between control and pulmonary hypertensive lung tissue specimens. At its current state, I think the paper has significant issues that need major reconsideration.
The authors obtained pulmonary tissue from a very small subset and performed cation exchange chromatography and LC/MS MS and analysed data using bioinformatics tools. Given how small the dataset are, the variability in gender between 'control' and experimental and the fact that 'controls' suffered from lung cancer and lobectomy, strongly question the validity of the findings. Variability in gender is significant here since women are more predisposed to pulmonary hypertension than men.

Experimental design

There are lots of variables included in the dataset, such as gender and number. The dataset needs more subjects for a comprehensive bioinformatic analysis. Control subjects should be true controls since lung cancer could have an effect on protein expression as well.
Fig 2- some of the GO terms on the X axis are incomplete

Validity of the findings

Some protein markers found in the study have already been shown to be upregulated in pulmonary hypertension models- intergrin a1 (Umesh et al.) and AMPK (Evans et al). This paper could have corroborated their data had the authors included some experimental data from a larger set of human patients directly implicating the concerned proteins.

Additional comments

Minimal bioinformatics analysis on a small dataset without proper controls and any experimental data to support the hypothesis provides room for a lot of future work.

Reviewer 2 ·

Basic reporting

Please check the article for grammatical or typographic errors, so that the logic is understandable to a general international audience.
I have listed some general comments about the way the manuscript is structured.

Experimental design

I think a one/two line description about the features and/or reach of the GO, KEGG analyses would be helpful for readers who are not experts in the field.
I commend the authors on the vast no. of proteins screened. I felt like it would be easier for the reader if fold changes of some of the major DEPs being looked at were mentioned in the main text. This would make the study more appealing.

Validity of the findings

No comment

Additional comments

The authors have profiled protein expression in lung tissues of patients suffering from pulmonary hypertension. While I commend the authors for identifying potential targets for therapeutic interventions, I think the manuscript is structured in a poor way and it is sometimes hard to follow the logic, especially in the Discussion section. The authors identified genes that are upregulated and downregulated, but then solely concentrated on the ones that were upregulated without providing any justification for the preference. In general, I think the manuscript needs to be revised to be of relevance to a general international audience.
I am curious why there is a significant difference in ages of the test and control populations. Can the authors comment on why the difference is not important.
The authors make extensive use of LC-MS for identifying proteins. It would be helpful to show representative images on how these proteins are identified. Also, it would be helpful to detail a few salient features of the GO, KEGG analyses for the general audience.
Line 183: Can the authors provide example of the specific proteins being looked at, in each of the three different pools?
Line 196: Can the authors be more descriptive of the 10 KEGG signaling pathways? Can the authors comment on interplay between the identified pathways or are they divergent.
Line 220-241: I do not understand the significance of the paragraph. The key finding is that AMPK is upregulated and hence a potential drug target.
In general it would be nicer to have the fold change information for the genes AMPK, ITGA1 etc. in the main text.
Lines 256-274: I do not understand how all of the ancillary functions are related to muscle contraction. It would be helpful if the authors can concisely state the findings. I think in the present form all of the additional description conceals discussion on major findings.
Can the authors comment on what is already known about the field. It seems like some of the key findings of the manuscript have already been highlighted by Evans et al, Umesh et al, Zhang et al etc. I think such prior findings should be discussed in the introduction.
I do not see any new information been provided in Figs. 3 and 4. They seem to be similar to Fig. 2 recast into different axes. It would be nicer to have references added to the pathways in Figs. 7 and 8.
Line 53: familial means the disease is associated with family history. The sentence seems redundant and should be rephrased.
Lines 62, 204, 224 and 229 are hard to follow or redundant.
Line 178: indicates

Reviewer 3 ·

Basic reporting

In this paper, the author has analyzed lung tissue specimens from several patients with pulmonary hypertension to study the protein expression profile changes for providing potential new biomarkers. Overall the paper has a present a comprehensive analysis on the protein expression profile with adequate experimental methods. A few things need to be addressed to improve the overall quality of the paper.

Experimental design

See below

Validity of the findings

See below

Additional comments

1. The English of the paper needs to be examined by a native speaker to avoid some grammar mistakes
2. Some abbreviations, such as GO, KEGG and PPI, didn't have the full descriptions when these terms started to appear, although there were descriptions in later parts.
3. In figure 5, 6 and 7, the author can minimize the pathway maps to the part where has most changes, so that readers can get a more clear understanding

---

## Round 0.2 · accepted · Accept

All critiques were adequately addressed and the manuscript was revised accordingly. Therefore the amended version is acceptable now.

Reviewer 2 ·

Basic reporting

In the revised manuscript, the authors have addressed my major concerns
and the manuscript has been improved. I recommend acceptance and publication.

Experimental design

No comments

Validity of the findings

No comments

Reviewer 3 ·

Basic reporting

The quality of the paper has improved a lot and all the points in the feedback have been addressed carefully.

Experimental design

N/A

Validity of the findings

N/A

Additional comments

N/A